# Insights on Visual Representations for Embodied Navigation Tasks

## Abstract

Recent advances in deep reinforcement learning require a large amount of training data and generally result in representations that are often over specialized to the target task. In this work, we study the underlying potential causes for this specialization by measuring the similarity between representations trained on related, but distinct tasks. We use the recently proposed projection weighted Canonical Correlation Analysis (PWCCA)to examine the task dependence of visual representations learned across different embodied navigation tasks. Surprisingly, we find that slight differences in task have no measurable effect on the visual representation for both SqueezeNet and ResNet architectures. We then empirically demonstrate that visual representations learned on one task can be effectively transferred to a different task. Interestingly, we show that if the tasks constrain the agent to spatially disjoint parts of the environment, differences in representation emerge for SqueezeNet models but less-so for ResNets, suggesting that ResNets feature inductive biases which encourage more task-agnostic representations, even in the context of spatially separated tasks. We generalize our analysis to examine permutations of an environment and find, surprisingly, permutations of an environment also do not influence the visual representation. Our analysis provides insight on the overfitting of representations in RL and provides suggestions of how to design tasks that induce task-agnostic representations.

## 1 Introduction

Recent advancements in deep reinforcement learning (deep RL) have allowed for the creation of systems that are able to out-perform human experts on a variety of different games such as Chess, Go, Dota2, and Starcraft2 (Vinyals et al., 2019; Silver et al., 2017; Tian et al., 2019). These advances have heavily relied on sample-inefficient algorithms that require significant amounts of task-specific training episodes, making them computationally expensive to run. Furthermore, deep RL has been found to be capable of overfitting to the training task, even for complex problems (Zhang et al., 2018), or failures when the environment is altered (even if this, in turn, simplifies the task Ruderman et al. (2019)). These observations call into question whether representations learned with one training task will be reusable for novel tasks.

The generality and reuse-ability of representations is a desirable and powerful property as it allows knowledge to be transferred between tasks and can help alleviate a lack of data. In the regime of supervised learning, it is well known that deep neural networks are capable of overfitting on tasks and memorizing random labels (Zhang et al., 2016), making it reasonable to expect that representations would be highly tuned to their training task. However, many have shown that representations trained for one task perform well for other tasks, both as an initialization for fine-tuning and as a static feature-extractor (Girshick, 2015; Ren et al., 2015; Anderson et al., 2018b; Conneau et al., 2017; Kornblith et al., 2019). Resolving this discrepancy is an area of much debate and active research (Neyshabur et al., 2019; 2015; Keskar et al., 2017; Golowich et al., 2018; Arora et al., 2018; Morcos et al., 2018b).

Reusing representations provides a promising avenue for the emerging field of training virtual robots in simulation before transfer learned skills to reality. There have been a number of recent works proposing to train robots as Embodied Agents in simulated environments with the ultimate goal of transferring agents learned in simulation to reality (Gupta et al., 2017; Zhu et al., 2017; Anderson et al., 2018c; Das et al., 2018; Gordon et al., 2018; Wijmans et al., 2019; Savva et al., 2019). The ability to reuse representations for new tasks and in new environments is of particular concern to the

goal of transferring embodied agents from simulation to reality. Once in the real world, an agent should be capable of learning new tasks – such as finding new objects or handling new questions – and be able to cope with the non-stationarity of a changing world. Thus, we seek to answer the following question: *Do different embodied navigation tasks induce different visual representations?*

**Contributions.** We study our primary question in the context of the task of Object Navigation (ObjectNav), *e.g. 'Go to the X'*. We define two different embodied tasks by constructing disjoint splits of target objects, allowing us to understand the exact differences between our tasks. We then adapt the methodologies proposed in Raghu et al. (2017) and Morcos et al. (2018a) to examine the impact the task has on the visual representation. We first perform our experiments using SqueezeNet1.2 (Iandola et al., 2016) as parameter efficient networks would be a good choice for embodied agents deployed on real robots. We find that, surprisingly, differences in the task do not lead to a measurable effect on the visual representation. We leverage this knowledge to show that visual representations trained for one tasks are useful for learning another, and, surprisingly, allow for more sample efficient learning.

Next, we design a special case where the different tasks constrain the agent to spatially disjoint locations in the environment, such that the agents should explore different areas during training, resulting in different representations and providing insight on how task independent visual representations emerge. We demonstrate that this task dependence negatively impacts the ability to re-use the representation for new tasks.

We then consider how our choice of CNN impacted our findings by performing our analysis on a second CNN, a version of ResNet50 (He et al., 2016) modified to have a comparable number of parameters to SqueezeNet1.2, and find similar conclusions for the non-special case set of tasks. For the spatially disjoint tasks however, the modified ResNet50 learns reasonably task-agnostic representations, in contrast to our results for SqueezeNet, suggesting that ResNets contain inductive biases that encourage more task-agnostic representations.

Finally, to evaluate the extent to which these results are environment dependent, we generalize our analysis to transfer across multiple permutations of an environment and demonstrate representations learned in one permutation of an environment are effective for the other permutations.

## 2 RELATED WORK

**Representation analysis.** Analyzing the representations of deep neural networks has been the subject of many works. Initial works focused on analyzing individual neurons (Li et al., 2016; Zeiler & Fergus, 2014; Bau et al., 2017; Arpit et al., 2017; Morcos et al., 2018b). In this work, however, we examine the entire representation. Our closest related works, Raghu et al. (2017); Morcos et al. (2018a), propose methods to examine the entire representation of neural networks in the context of standard image classification tasks. We adopt their analysis tools and utilize them to analyze neural networks in the context of *embodied*-vision tasks and reinforcement learning. See Section 3.2 for a more detailed discussion of the benefits of these methods.

**Reward-free reinforcement learning.** Transfer of knowledge and representations is a paradigm commonly used in task-agnostic and reward-free reinforcement learning. The goal of this paradigm is to allow the agent to interact with its environment such that it gains general knowledge, thereby allowing it to learn downstream tasks with less samples. These works provide the agent with a reward signal such that it will explore its environment (or state-space). This can be formulated from an information theoretic standpoint to provide intrinsic motivation Jung et al. (2011); Eysenbach et al. (2018); Gregor et al. (2016); Haarnoja et al. (2017). Others provide a more direct signal in the form of exploration based rewards Burda et al. (2018); Savinov et al. (2018). We differ from these works by using representations learned via task-driven reinforcement learning directly for a different task.

**Transfer Learning.** Transfer learning seeks to transfer knowledge between a domain with labeled data to another domain (Pan & Yang, 2009; Luo et al., 2017; Oquab et al., 2014). Transfer learning has also been studied in the context of reinforcement learning by designing specific objectives or model structures such that knowledge can be transferred between two tasks Oh et al. (2017); Bacon et al. (2017); Taylor & Stone (2009). We do not use any specific architecture or objectives and examine task dependence of vanilla architectures.

## 3 APPROACH

In this section, we outline our approach for answering our core question by describing the task we examine and the method for comparing representations we leverage.

### 3.1 EXPERIMENTAL SETUP

**Task.** We examine the task of Object Goal Navigation (ObjectNav) due to its reliance on both semantic and spatial understanding. In ObjectNav, an agent is given a token describing an object in the environment, such as *fridge*, and then must navigate through the environment until it finds a good view of the fridge and calls the stop action. To avoid under-specification of the task, we restrict target objects to have at most two instances for a given class. Note that each target object is specified uniquely by its object ID. The terminal reward given is proportional to how much of the target object is in the agent's field of view. At every time-step, a shaped reward proportional to the agents progress towards the target object is also provided. See the supplementary for more details.

**Environment.** We use the extreme high-fidelity reconstructions in the Replica Dataset (Straub et al., 2019) and simlate agents utilizing AI Habitat (Savva et al., 2019). We utilize these environment so that our analysis will be more applicable to the ultimate goal of agents operating in reality. See Fig. 1a for a top-down view of an environment and the supplement for example agent views.

**Agent.** The agent has 4 primitive actions, move_forward, which moves $0.25$ meters forward; turn_left and turn_right (which turn $10$ degrees left and right, respectively), and stop which signals that the agent believes it has completed its task. At every time-step, the agent receives an egocentric RGB image and the token specifying the target object.

**Policy.** We parameterize our agent with 3 components. A visual encoder, a target encoder, and a recurrent policy. The visual encoder utilizes SqueezeNet1.2 (Iandola et al., 2016) as the backbone architecture as its combination of parameter efficiency and representational power is a logical choice for embodied agents deployed on real robots. The target encoding is a $128$ dimensional vector that is learn-able and unique for each target object. The policy consists of a GRU (Cho et al., 2014) followed by 2 fully connected layers. See the supplementary for more details. Note that the vast majority ($\sim$80%) of the learnable parameters are in the visual encoder. This is key to our analysis as other the network is able to perform the task with a frozen randomly initialized visual encoder.

**Training.** We use Proximal Policy Optimization (PPO) (Schulman et al., 2017) with Generalized Advantage Estimation (Schulman et al., 2015) and the Adam optimizer (Kingma & Ba, 2014) to train our agent. We train for 15,000 rollouts ($\sim 61 \times 10^6$ steps) to ensure converge across different random seeds. See the supplementary for more details.

### 3.2 MEASURING THE SIMILARITY OF REPRESENTATIONS

A perhaps straight-forward approach to measuring the similarity of representations would be to simply measure the distance (e.g., Euclidean or cosine) between their representations of the same inputs. However, this approach is ill-suited to neural networks. Consider the following toy example: For a set of inputs $\mathcal{X}$, suppose that function $f$ produces a representation that is uniform on the N-ball and define $f' = Af$ for an affine transform $A$. A simple distance calculation (or alternatively, dimensionality reduction and clustering) would report a high distance between the two representations. Accounting for affine transformations is important when analyzing neural networks as, for any given layer, one can apply any affine transformation to the activations and the inverse to the next layer's weights without changing the network. Given two neural networks trained in the exact same way modulo the random seed, there is no reason why their representations would be aligned despite computing very similar (if not exact the same) functions (Li et al., 2016).

Instead, we follow the approach of Raghu et al. (2017); Morcos et al. (2018a) to compare the representations of two deep neural networks. Given two neural networks, A and B, and a set of $N$ inputs, Raghu et al. (2017); Morcos et al. (2018a) compare the representations at layer $L$ of both networks by 1) extracting the neuron activation matrix, $X$, of both networks – where $X_{i,j}$ is the activation of the $i^{th}$ neuron on the $j^{th}$ input; and 2) compute the distance between the neuron activation matrices using Canonical Correlation Analysis (CCA), a classic statistical technique (Hotelling, 1936). CCA finds a basis which maximizes the correlation between two matrices and then computes the correlation in that basis, thereby account for any affine transformations between two representations.

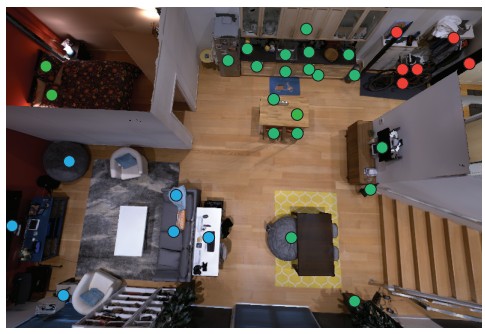

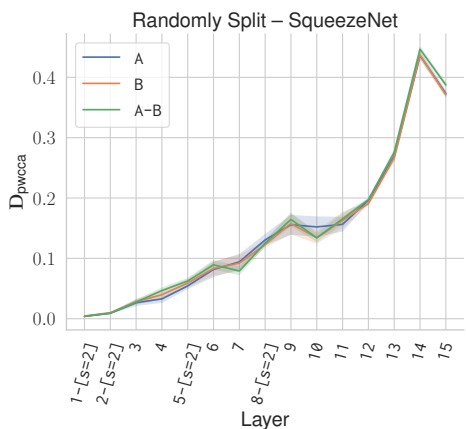

(a) Top-down view in the environment. Circles denote the location of all target objects. Coloring denotes which target set objects are in for the spatially disjoint split: blue for $\mathcal{A}$, red for $\mathcal{B}$, and green is unused.

(b) PWCCA results of comparing networks trained on different embodied task. Down-sampling layers are marked with [s=2]. Shading around the line corresponds to a $95\%$ confidence interval calculated via empirical bootstrapping.

Figure 1

It is worth noting that CCA (and variants) do not capture the "usefulness" of representation to the downstream task.

We follow the technique proposed by Morcos et al. (2018a) to account for differing numbers of noise dimensions between representations. This method weights CCA correlation coefficients by the amount of variance each CCA direction explains in the real data. Given each of the CCA directions $h_i$ and correlation coefficients $\rho_i$, Morcos et al. (2018a) first computes the projection coefficients $\alpha_i = \sum_k |\langle d_i, X_k \rangle|$ and then computes 1 minus the weighted average of the correlation coefficients, $\mathbf{D}_{\text{pwcca}} = 1.0 - \frac{1}{\sum_k \alpha_k} \sum_k \alpha_k \rho_k$, as the distance between representations.

## 4 HOW TASK-DEPDENDENT ARE LEARNED REPRESENTATIONS?

**Core Hypothesis.** Training for different embodied tasks induces different visual representations. Due to Deep RL's ability to overfit on even complicated tasks, it is reasonable to expect that the representations learned will be highly tuned to their specific task.

**Two tasks.** To gain insight into the impact of task differences on visual representations, we must first understand the differences between the tasks themselves. An ideal task set should contain tasks for which the learning and reward dynamics are very similar, but which differ in simple and easily understandable ways. To accomplish this, we randomly divide the set of target objects, $\mathcal{X}$, into two equally sized and disjoint subsets $\mathcal{A}$ and $\mathcal{B}$ such that $\mathcal{A} \cap \mathcal{B} = \varnothing$, $\mathcal{A} \cup \mathcal{B} = \mathcal{X}$, and $|\mathcal{A}| = |\mathcal{B}|$ (assuming $|\mathcal{X}|$ is even). We average our results over three different choices of $\mathcal{A}$ and $\mathcal{B}$. These two tasks therefore share the same environment and action space and have similar visual statistics, but differ only in the set of target objects to which the agent must navigate. To control for the effect of any particular environment, we rerun these results over 4 additional environments in the Replica dataset – apartment_0, office_2, room_0, frl_apartment_0.

**Measuring the task dependence of representations.** A naive approach to using PWCCA to measuring the effect of different target sets on the representation learned would be to train a policy for $\mathcal{A}$ and a policy for $\mathcal{B}$ and then measure the dissimilarity. This approach doesn't control for the effect of different random initialization, and, more importantly, doesn't ground the values reported by PWCCA (which is a unitless metric). Instead, we compare the distance between models trained on *different* tasks to the distance between models trained on the *same* task. If the distance between models trained on different tasks is *higher* than that between models trained on the same task, representations are *task-dependent* whereas if the distance between models trained on different tasks is the *same* as that between models trained on the same task, representations are *task-agnostic*.

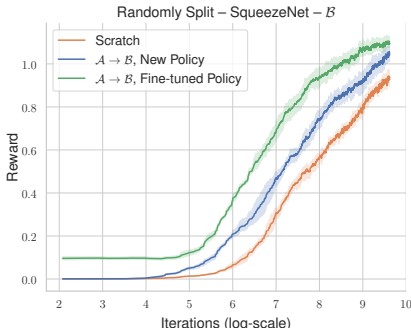 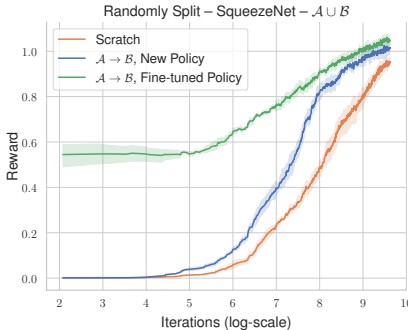

Figure 2: Results of transferring policies learned on one task to the other task. Train reward while learning target set $\mathcal{B}$ (left) and target set $\mathcal{A} \cup \mathcal{B}$ (right) under three different regimes.

To compare representations across different tasks, we train N networks for $\mathcal{A}$ and N networks for $\mathcal{B}$, compute the PWCCA distance for each pair of networks, and then average over the $N^2$ pairwise comparisons. To compare representations learned for the same task, we take the N networks trained on $\mathcal{A}$ (or $\mathcal{B}$), and compute the PWCCA distance for the $\binom{N}{2}$ network pairs.

We use the following notation to denote our comparison: comparisons across networks trained on the same task are denoted without a dash, *e.g.* A is the comparison of networks trained on $\mathcal{A}$ among themselves. Comparisons across networks trained on different tasks are denoted with a dash, *e.g.* A-B is the comparison *between* networks trained on $\mathcal{A}$ and networks trained on $\mathcal{B}$.

**Representations are not influenced by the training task.** If networks trained on different tasks learn different representations, we would expect the A-B distance to be higher than that for A or B alone. In contrast, we found that distances were similar regardless of task trained, suggesting that networks learn task-agnostic visual representations, Fig. 1b. This result is surprising as it implies that the differences in learning dynamics, reward, and incentives induced by the different target splits *have no more impact on the representation than the random seed alone*. Despite arising directly from training on that set of target object, the visual representation shows no bias in how it represents the environment. To determine whether this effect is dependent on the particular environment used, we repeated this analysis across four additional environments and found similar trends (Fig. A3). A direct and actionable implication of this result is that the representation learned for one task should transfer to another.

## 5 Transferring between $\mathcal{A}$ and $\mathcal{B}$

We aim to generate policies with task agnostic visual representations as we hope that these visual representations can be easily adapted to new tasks. In this section, we evaluate whether the PWCCA results above, which suggest that agents learn task-agnostic representations, also imply that representations learned on $\mathcal{A}$ are sufficient to learn $\mathcal{B}$.

**Setup.** We examine two types of transfer experiments: 1) transferring the policy learned on $\mathcal{A}$ to $\mathcal{B}$ (or from $\mathcal{B}$ to $\mathcal{A}$), and 2) transferring the policy learned on $\mathcal{A}$ to $\mathcal{A} \cup \mathcal{B}$ (the full set of targets). In *all* transfer experiments, every layer of the visual encoder is frozen. We consider both fine-tuning the policy learned on $\mathcal{A}$ and learning a new policy from scratch.

**Results.** As suggested by the PWCCA experiments, we found that visual representations learned on $\mathcal{A}$ are effective for learning both $\mathcal{B}$ and $\mathcal{A} \cup \mathcal{B}$ (Fig. 2). We also found fine-tuning to be more effective than learning a new policy from scratch, suggesting that general navigation skills can transfer in addition to visual representations. These results suggest that the representational similarity observed in Sec. 4 leads to directly transferable representations, and confirms that agents in this environment learn task-agnostic representations.

**Sample efficient learning of new target objects.** We also consider the sample efficiency of learning $\mathcal{B}$ with a *frozen* visual representation trained for $\mathcal{A}$ compared to learning $\mathcal{B}$ from scratch. Imagine an agent deployed as a home robot: it can be pre-trained for some set of target objects but then must be capable of learning new objects over time. Ideally we would be able to share and re-use large parts of the agent – its visual encoder for instance – to learn these new target objects. The results in Fig. 2

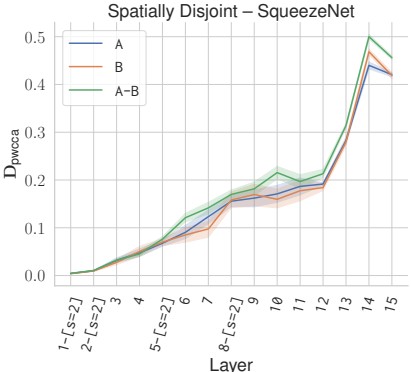 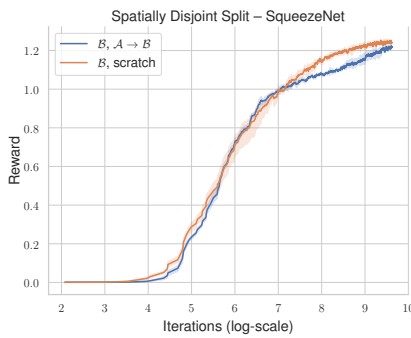

Figure 3: (Left) PWCCA analysis on the spatially disjoint target sets for SqueezeNet. (Right) Transfer results on the spatially disjoint split for SqueezeNet

(left) imply that using the representation learned on $\mathcal{A}$ as a feature-extractor may provide an efficient method for learning new sets of target objects.

We measure sample efficiency by training with five different random seeds and recording the number of iterations (rollouts) needed to reach a reward of $0.8$ on average, which represents good performance on this task. We compare learning $\mathcal{B}$ from scratch, and learning $\mathcal{B}$ with a *frozen* visual representation pre-trained on $\mathcal{A}$. While Fig. 2 (left) shows that fine-tuning a copy of the existing policy is more sample efficient, we examine the more general case of learning the policy from scratch.

Surprisingly, utilizing a frozen visual representation learned on $\mathcal{A}$ is a more sample efficient strategy for learning $\mathcal{B}$ than learning $\mathcal{B}$ from scratch (Fig. 2), suggesting that the visual represnetation learned on $\mathcal{A}$ is generalizable. We note that learning $\mathcal{B}$ could potentially be done more efficiently as we have not optimized our selection of reinforcement learning algorithm for sample efficiency. An additional benefit of re-using and freezing the visual encoder is that reinforcement learning algorithms which provide increased sample efficiency but have difficulties scaling to millions of parameters can be used instead of PPO (which scales to millions of parameters, but does not maximize sample efficiency).

## 6   A SPATIALLY DISJOINT SPLIT

In the previous sections, we demonstrated that the visual representations learned across tasks are highly similar and can be transferred across tasks, but the aspects of these tasks which enable task-agnostic learning remain unclear. One possibility is that both target sets cover the entire visual manifold, leading agents to explore the same portions of the environment across tasks. To test this hypothesis, we created hand-designed target sets which contain little to no spatial overlap.

**Setup.** We examine spatially disjoint sets with multiple target objects (see Fig. 1a (red vs. blue)). We examine the effect of this spatially disjoint split on both the PWCCA results and the transfer results.

**Representation Similarity.** If the underlying factor that causes representations to be similar is spatial coverage, we would expect the distance between the representations learned on $\mathcal{A}$ and the representations learned on $\mathcal{B}$ to increase in the spatially disjoint case. Consistent with this hypothesis, we found that representations differed between $\mathcal{A}$ and $\mathcal{B}$ when models were trained on spatially disjoint splits (Fig. 3). This result suggests that our previous PWCCA results (Fig. 2) were not merely finding similarity were not artifactual and provides support for our hypothesis that similar coverage is a necessary condition for representational overlap. We repeat this analysis over four additional environments and find similar trends (see Fig. A5).

**Representation re-usability.** The PWCCA results show that agents trained on the spatially disjoint split learn different representations for $\mathcal{A}$ and $\mathcal{B}$. We also examine if this difference in representation affects the re-usability of representations. We do so by re-running the transfer efficiency experiments[1]. We found that while $\mathcal{B}$ can still be learned with a representation trained on $\mathcal{A}$ (Fig. 3), we no longer observe the gain in sample efficiency observed on the randomly divided target object splits, confirming that the learned representations are more distinct. We note, however, that transferred

---

[1]note that the representation learned on $\mathcal{A}$ is taken after 2,000 iterations (rollouts) of training for $\mathcal{A}$

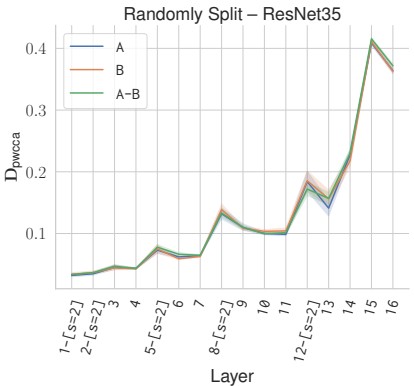
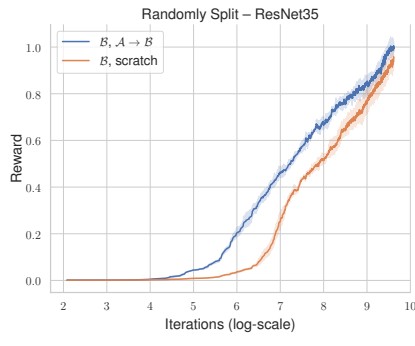

Figure 4: (Left) PWCCA results of comparing networks trained on different embodied task for the ResNet35 model. (Right) Transfer results for randomly split target objects for the ResNet35 model.

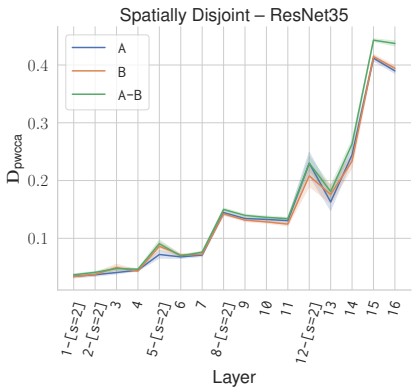
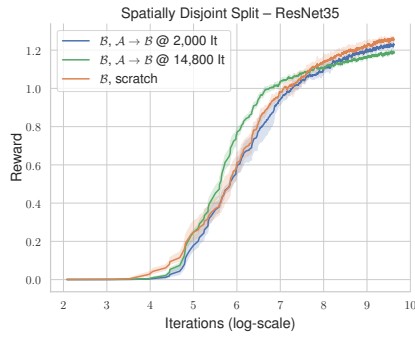

Figure 5: (Left) PWCCA results of comparing networks trained on the spatially disjoint split for ResNet35. (Right) Transfer results for disjoint split of target objects for the ResNet35 model.

representations are not substantially worse than representations learned from scratch, suggesting that some information is shared even in the spatially disjoint case.

## 7 A DIFFERENT ARCHITECTURE

Next, we test how these trends transfer to a different architecture. Specifically, we examine a modified ResNet50 (He et al., 2016) architecture. We reduce the number of parameters such that the network has a similar number of parameters to SqueezeNet1.2. The resultant network has 35 layers, and we therefore refer to it as ResNet35. We replace all Batch Normalization layers with Group Normalization (Wu & He, 2018) layers. See the supplementary material for more details. We train with the previous procedure and hyper-parameters.

**Random target splits.** Consistent with SqueezeNet models, we found that randomly distributed target objects have no effect on the visual representation learned (Fig. 4), indicating that our initial choice of CNN had no impact on this result. We repeat this analysis over four additional environments and find similar trends (see Fig. A4). We also observe an interesting behavior between the down-sampling layers; the distance between representations induced by different random seeds decreases. This suggests that residual connections help networks learn more similar representations. Fig. 4 shows the results of using a representation learned on $\mathcal{A}$ to learn $\mathcal{B}$. We once again see that the representation learned on $\mathcal{A}$ is sufficient for learning $\mathcal{B}$ and that this is a more sample efficient strategy than training a network for $\mathcal{B}$ from scratch.

**Spatially disjoint split.** In contrast to our results on SqueezeNet, we observed qualitatively different behavior in ResNet35 models trained on spatially disjoint splits. While spatially disjoint splits induced different learned representations in Squeezenet, for ResNet35, we observed minimal separation between the inter-comparisons (A–B) and the intra-comparisons (A, B), see Fig. 5. We repeat this analysis over four additional environments and find similar trends (see Fig. A6).

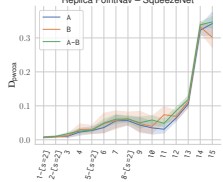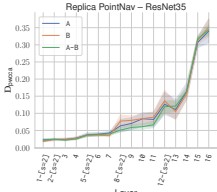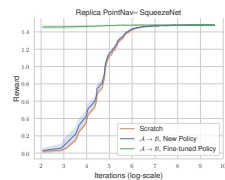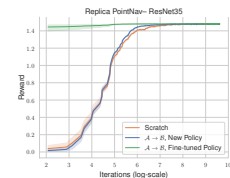

Figure 6: Our main experiments (PWCCA and transfer-ability) for disjoint sets of training *environments*. We find results consistent with our prior findings – singular differences to not impact the representation in a measurable way.

We then transferred spatially disjoint representations, and found that learning $\mathcal{B}$ from the representation learned on $\mathcal{A}$ after 14,800 iterations (the same number of iterations as the PWCCA plot) is faster than learning from scratch (Fig. 5). However, if we take the representation after 2,000 iterations (when the average reward on $\mathcal{A}$ reaches 0.8), we observe the reverse, implying that after 2,000 iterations the representation learned on $\mathcal{A}$ is specific to that task, but converges to a general one.

## 8 GENERALIZATION TO MULTIPLE ENVIRONMENTS

Finally, we generalize our analysis to multiple permutations of an environments. Specifically, the Replica dataset contains 6 different version of the same apartment, with dramatically different configurations of the objects (see `frl_apartment_{0-5}`). We look at the question *Does the representation depend on the position of objects?*. We examine this question by training agents for PointGoal Navigation (Anderson et al., 2018a) – in PointNav an agent must navigate to a given location specified by a point in ego-centric coordinates – where $\mathcal{A}$ is a random selection of 3 environments and $\mathcal{B}$ is the remaining. We once again find that the representation learned is surprisingly invariant to changes (Fig. 6) – the location of objects does *not* impact the representation in a measurable way. We verify this with transfer experiments and find that, in this case, the information learned in environments $\mathcal{A}$ is sufficient to perform the task well in environments $\mathcal{B}$.

## 9 DISCUSSION

We present a series of results and analysis centered around the question: *Do different embodied navigation tasks induce different visual representations?* To answer this question, we constructed two embodied navigation tasks by creating disjoint splits of target objects for the task of ObjectNav. We then used PWCCA (Raghu et al., 2017; Morcos et al., 2018a) to measure the influence of the task on the representation, the first to do so for deep RL. We found that for both SqueezeNet and ResNet visual encoders, the task does not influence visual representation, allowing for use in learning new tasks in a sample efficient manner. We then hand-designed a spatially disjoint split to create tasks than influence the visual representation. We found that this has the desired effect for our SqueezeNet models, but does not for the ResNet models. Our work provides valuable and actionable insight into how the task influences the representation for embodied navigation tasks.

**Caveats.** Our results and analysis have the following primary caveat. The two tasks we examine, while distinct, are quite similar. Designing experiments for this type of analysis across tasks with less similarity while not introducing too many additional variables is an avenue for future work.

**Takeaways.** We show that under certain settings, task agnostic visual representation can be induced. Our results suggest that on ingredient is coverage of the visual space that will be seen, implying that designing tasks and environments which maximize the visual diversity seen by the agent is paramount. However, this conclusion is somewhat less clear following our experiments with ResNet35, which appear to learn more task-agnostic representations in general. While ResNets have been used for embodied vision tasks in prior works (Fang et al., 2019; Sax et al., 2018; Anderson et al., 2018c; Wijmans et al., 2019), they are not common and and are rarely trained directly for their task. Our results suggest that utilizing ResNets, even with dramatically less parameters, will help to transfer representations between different embodied tasks.

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

# A    ARCHITECTURE DETAILS

## A.1    SQUEEZENET ENCODER

For our SqueezeNet1.2 (Iandola et al., 2016) based visual encoder, we utilize all layers expect for the final convolution and global average pool. Given a 224×224 image, this produces a (512×13×13) feature map. We follow this with two convolution layers, `Conv-[42, k=3, d=2]`, `Conv-[21, k=3, d=1]` where d specifies the dilation, to produce a (21×7×7) feature map. This feature map is then flattened and transformed to a 256d vector with a fully connected layer.

## A.2    RESNET ENCODER

For our ResNet50 (He et al., 2016) based visual encoder, we start with all residual layers (all layers minus the global average pool and softmax classifier). We then reduce the number of output channels at each layer by a factor of 4. We then remove 1 residual block within each layer and remove an additional residual block in the 3rd layer. ResNet50 contains 3 blocks in the first layer, 4 in the second, 6 in the third, and 3 in fourth. Our ResNet35 contains 2 blocks in the first layer, 3 in the second, 4 in the third, and 2 in the fourth. We replace all Batch Normalization layers with Group Normalization (Wu & He, 2018) layers, to account for the highly correlated observations seen in on-policy reinforcement learning.

We reduce the 512×7×7 feature map to 41×5×5 with two convolution layers, `Conv-[41, k=1]`, `Conv-[41, k=3]`. This feature map is then flattened and transformed to a 256d vector with a fully connected layer.

## A.3    POLICY

Given the 256-d visual feature, we concatenate the 128-d target encoding and use the resulting 384-d vector as input to a single layer GRU (Cho et al., 2014) with a 256-d hidden state. The hidden state is reduced to 128-d with a fully connected layer, and the 128-d representation is used to produced the softmax distribution over the action space and estimate the value function.

# B    IMPLEMENTATION DETAILS

We utilize an in-house built simulator to perform our experiments. We perform collision checking on a pre-computed occupancy grid and no partial steps are allowed.

Models are trained on a single node with 8 Tesla V100 GPUs. We use PyTorch to trian our agent; we base our training code on (Kostrikov, 2018).

We utilize the publicly available implementation of PWCAA (Morcos et al., 2018a): https://github.com/google/svcca

# C    TASK DETAILS

The reward at time $t$ is given as follows:

$$R_t = \begin{cases} \frac{\mathbf{IoU_t}}{\mathbf{IoU}_{\max}} & \text{action} = \texttt{stop} \\ -0.05 \cdot \Delta_{\text{geo\_dist}} & \text{otherwise} \end{cases}$$

Where $\mathbf{IoU}$ is the intersection over union between the semantic segmentation of the target object and a predefined bounding box. $\mathbf{IoU}_t$ is the $\mathbf{IoU}$ at the agents current position. $\mathbf{IoU}_{\max}$ is the maximum possible $\mathbf{IoU}$ for the target object as determined by exhaustive search within a reasonable radius of the target object.

# D    TRAINING DETAILS

We use Proximal Policy Optimization (PPO) (Schulman et al., 2017) with Generalized Advantage Estimation (Schulman et al., 2015). We set the discount factor, $\gamma$, to 0.99 and $\tau$ to 0.95. We collect 128 frames of experience from 32 agents running in parallel (possibly working on different tasks) and then perform 4 epochs of PPO with 2 mini-batches per epoch. We utilize the Adam optimizer (Kingma

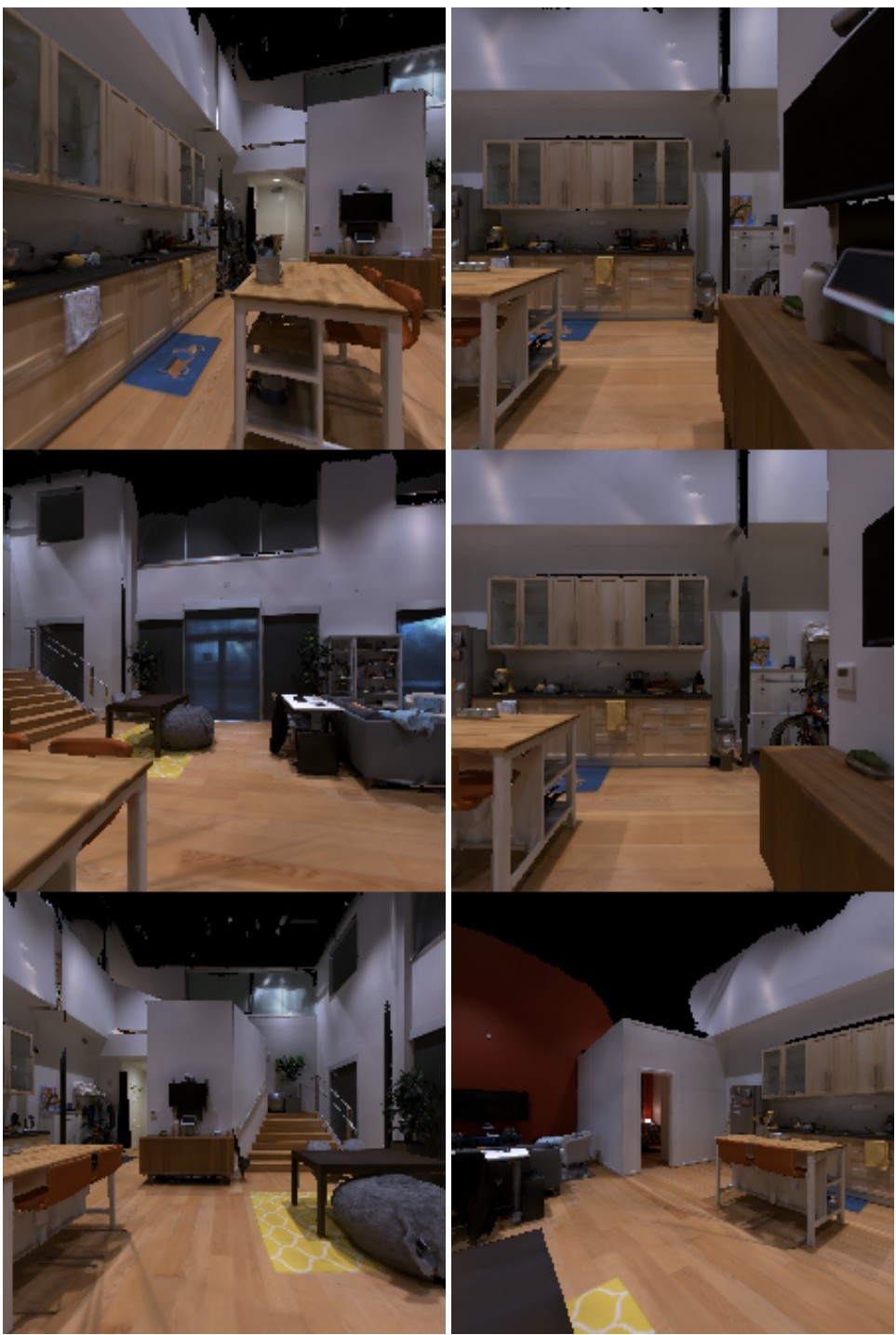

Figure A1: Example images from the environment we utilize.

& Ba, 2014) with a learning rate of $10^{-4}$ and a weight decay of $10^{-5}$. Note that unlike popular implementations of PPO, we do not normalize advantages as we find this often leads to instabilities during training. We train for 15,000 rollouts ($\sim 61 \times 10^{6}$) to ensure converge across different random seeds.

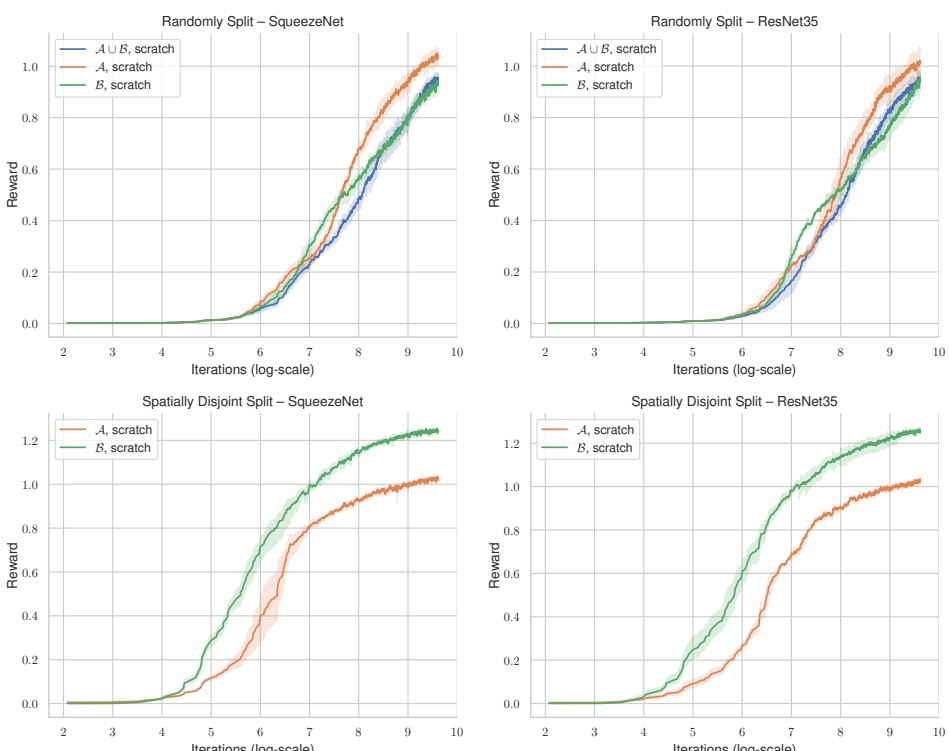

Figure A2: Reward curves for both models on various sets of tasks.

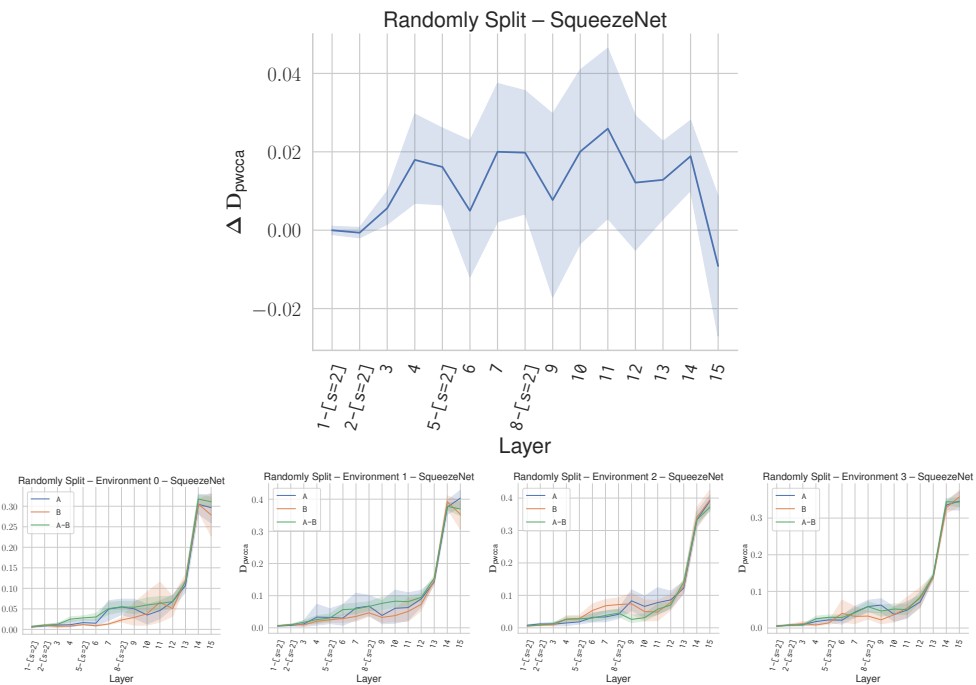

Figure A3: SqueezeNet results for randomly split sets of target objects on 4 environments from the replica dataset. First plot shows the average $\Delta\mathbf{D}_{\text{pwcca}} = \mathbf{D}_{\text{pwcca}}(\text{A-B}) - (\mathbf{D}_{\text{pwcca}}(\text{A}) - \mathbf{D}_{\text{pwcca}}(\text{B}))/2.0$ across all environments.

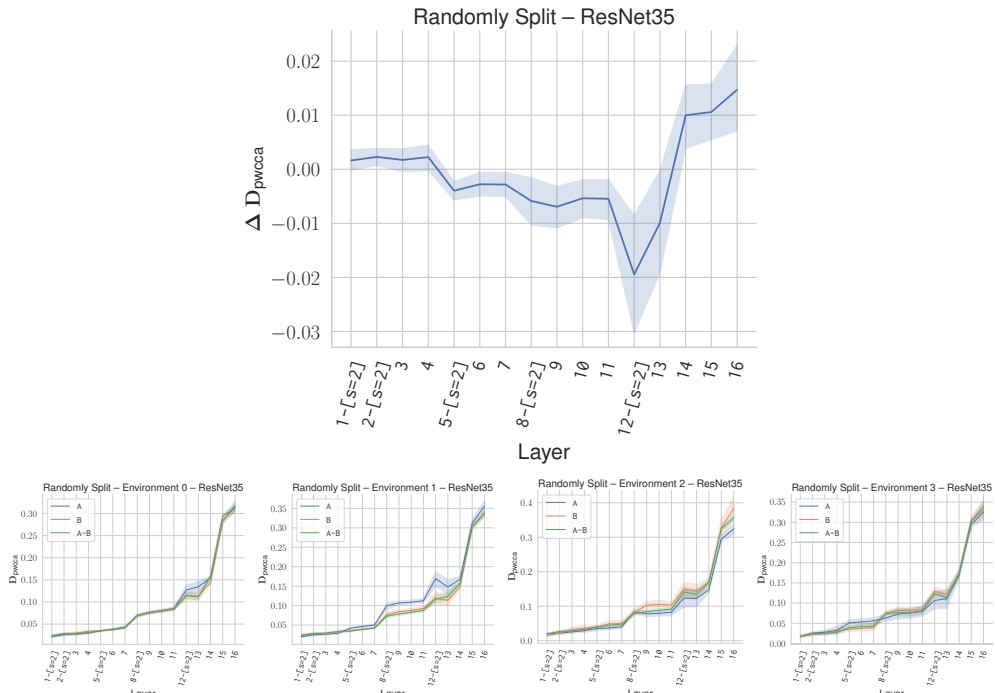

Figure A4: ResNet35 results for randomly split sets of target objects on 4 environments from the replica dataset. First plot shows the average $\mathbf{\Delta D}_{\text{pwcca}} = \mathbf{D}_{\text{pwcca}}(\text{A-B}) - (\mathbf{D}_{\text{pwcca}}(\text{A}) - \mathbf{D}_{\text{pwcca}}(\text{B}))/2.0$ across all environments.

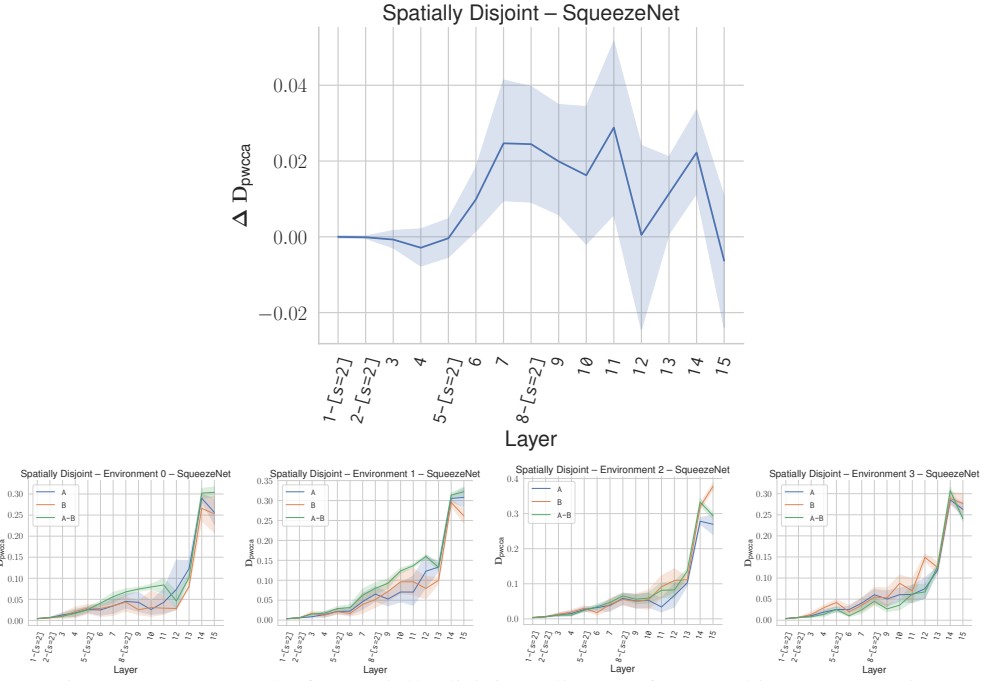

Figure A5: SqueezeNet results for spatially disjoint split sets of target objects on 4 environments from the replica dataset. First plot shows the average $\mathbf{\Delta D}_{\text{pwcca}} = \mathbf{D}_{\text{pwcca}}(\text{A-B}) - (\mathbf{D}_{\text{pwcca}}(\text{A}) - \mathbf{D}_{\text{pwcca}}(\text{B}))/2.0$ across all environments.

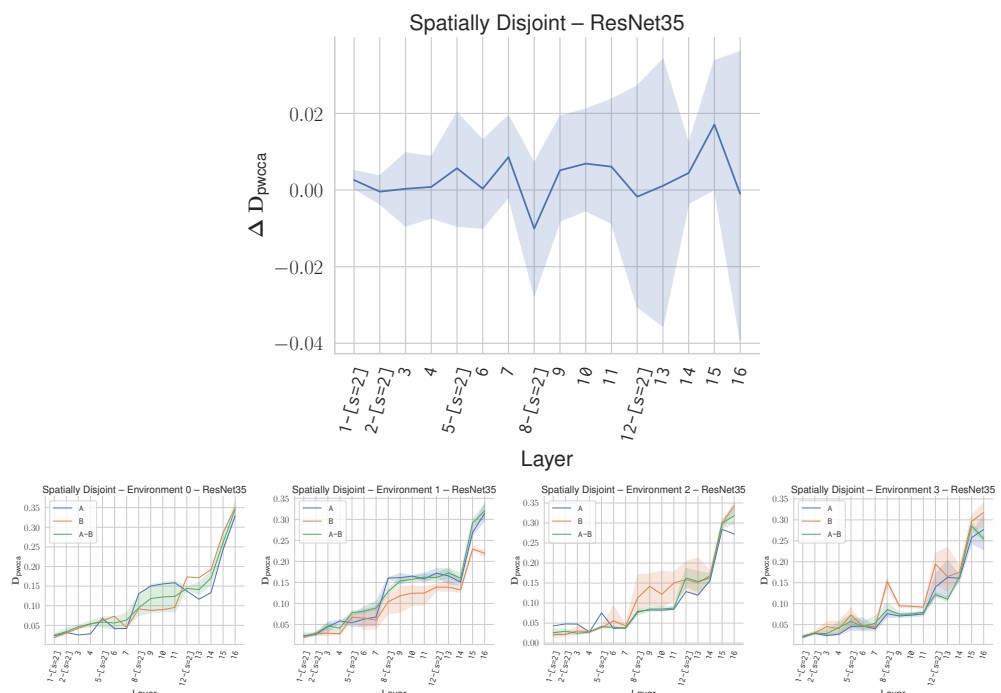

Figure A6: SqueezeNet results for spatially disjoint split sets of target objects on 4 environments from the replica dataset. First plot shows the average $\mathbf{\Delta D}_{\text{pwcca}} = \mathbf{D}_{\text{pwcca}}(\text{A-B}) - (\mathbf{D}_{\text{pwcca}}(\text{A}) - \mathbf{D}_{\text{pwcca}}(\text{B}))/2.0$ across all environments.

