# OpenReview forum: "Insights on Visual Representations for Embodied Navigation Tasks"
_ICLR.cc/2020/Conference — Reject_

### Official Review · AnonReviewer3 · 2019-10-20
**Official Blind Review #3**

**Rating:** 3

**Review:**

This paper tests generality and transfer in visual navigation tasks. It's an interesting question and a well-executed study. I applaud the use of a complex high-dimensional environment.

The experiments are well done, but I am not sure what we learn. Each experiment compares two highly similar tasks - as the authors themselves acknowledge - in ways that do not obviously connect to realistic transfer scenarios, such as transferring to a new environment. I would have liked to see experiments that compare networks trained on different environments or on different sets of environments.

The paper is called "Insights on visual representations for embodied navigation tasks," but I am left wondering what those insights are. The authors state that "Our work provides valuable and actionable insight into how the task influences the representation for embodied navigation tasks," but it is light on specifics and on the general discussion of the results. Not much is offered beyond a suggestion to use ResNets. I would consider revising my assessment if the authors write a more thorough discussion and specify clear implications of their findings.

The paper is generally well-written although important details are left out:
- I think the target objects have fixed rather than randomized locations in the environments, but this isn't stated in the paper. Please specify.
- Where does the agent start during an episode?
- How large are the environments? e.g., one room or multiple rooms?
- How many objects per environment are there e.g., in sets A and B?
- The "permutations of an environment" setting isn't very clearly written and it took me a few readings to understand what you mean.
- Finally, I suggest revising the very vague title to the paper

**Experience Assessment:**

I have published one or two papers in this area.

**Review Assessment: Checking Correctness Of Derivations And Theory:**

N/A

**Review Assessment: Checking Correctness Of Experiments:**

I carefully checked the experiments.

**Review Assessment: Thoroughness In Paper Reading:**

I read the paper at least twice and used my best judgement in assessing the paper.

---

> ### Author Response · Authors · 2019-11-15
> **Response to R3**
>
> > . I would have liked to see experiments that compare networks trained on different environments or on different sets of environments.
>
> We agree that extending these experiments to a more diverse set of environments presents and interesting avenue for future work.  In this work, we chose to examine a constrained set of environments such that all the differences between given tasks could be understood.
>
> > The paper is called "Insights on visual representations for embodied navigation tasks," but I am left wondering what those insights are......
>
> We agree.  Our experiments examine an interesting question and end up drawing a surprising and somewhat non-intuitive result that seems to suggest that task agnostic representations arise in networks by overlap in experience during training and this experience can be accrued while navigating to arbitrary goals.  However, they do not provide deep insight into why this may be true.  We hope this work will inspire others to help us dig deeper into this question.
>
> > Specifications of task and environments
>
> The target objects have fixed locations.  The agent is spawned in a random location at most 6 meters away from the target object.  Most environments are one or two rooms, one environment is a two story house. The number of target objects in A and B varies between 6 and 20 depending on the environment.  We will add these details to the paper.
>
>
> > The "permutations of an environment" setting isn't very clearly written and it took me a few readings to understand what you mean.
>
> We will clarify this section.
>
> > Finally, I suggest revising the very vague title to the paper
>
> Agreed.  We will revise the title to “Examining the Task Dependence of Visual Representations in Embodied Navigation”

---

### Official Review · AnonReviewer1 · 2019-10-22
**Official Blind Review #1**

**Rating:** 3

**Review:**

This article studies the similarities between the learned representations for different tasks when trained using reinforcement learning algorithms. The ultimate question that this study tries to answer is an interesting one. Namely, how much can representations learned by training on one task be beneficial for learning other tasks? A high interdependence between the representations can lead to a more successful transfer of knowledge between tasks.
The authors are further interested in studying the properties that influence this relationship, which depends on the elements of training as well as attributes of the tasks themselves.

However, I believe that the results are not strong enough to support the claims that are stated in the paper and the limited scope of the environments tested does not make a convincing case that the results will be generalizable much beyond these scenarios. Therefore, in the current state, I think this paper should be rejected.

Firstly, the paper states that:
> ".. if the distance between models trained on different tasks is the same as that between models trained on the same task, representations are task-agnostic."
This seems to be a key argument in the paper. However, I believe that this argument is based on the assumption that representations learned for a single task are indeed highly similar. I think this assumption requires some sort of support as reinforcement learning algorithms are known to be highly inconsistent even in reaching similar solutions. Therefore, one cannot take for granted that the learned representations would be similar in any way.

Second, the authors claim in Section 5 that the random splits have little impact on the learned representation, but in Section 6 claim that the spatially disjoint splits have a noticeable impact on the representations. Without any measure of the impact, looking at figures 1b and 3a, I'm not convinced that this distinction is so obvious. The situation is worse when looking at the figures in the appendix, namely figures A3 and A5. If someone were to swap these two figures, my untrained eye would not be able to tell the difference.

I suggest that the authors spend more time explaining their reasoning as to why these results are significant enough to support the claims. Also, the text should be improved if it is to be accepted. There exist many problems ranging from small typos (simlate -> simulate, reuse-ability -> reusability, and "?." -> "?") to sentences that need to be reworked. Some figure legends/captions can also be improved to include more information, such as explaining what exactly the shaded regions represent (I'm guessing one standard deviation from the mean over some unknown replicates), or in Fig 3b making clear whether "A -> B" is done with "New Policy" or "Fine-tuned Policy". I am also curious as to why only one of these scenarios was experimented with in sections 6 and 7.

**Experience Assessment:**

I have read many papers in this area.

**Review Assessment: Checking Correctness Of Derivations And Theory:**

I did not assess the derivations or theory.

**Review Assessment: Checking Correctness Of Experiments:**

I assessed the sensibility of the experiments.

**Review Assessment: Thoroughness In Paper Reading:**

I read the paper at least twice and used my best judgement in assessing the paper.

---

> ### Author Response · Authors · 2019-11-15
> **Response to R1**
>
> > This seems to be a key argument in the paper. However, I believe that this argument is based on the assumption that representations learned for a single task are indeed highly similar. I think this assumption requires some sort of support as reinforcement learning algorithms are known to be highly inconsistent even in reaching similar solutions. Therefore, one cannot take for granted that the learned representations would be similar in any way.
>
> Our analysis does not need the models to be identical, but to share a subspace in their representations.  While it is possible that every model could learn representations with no common subspace, our experiments show that models trained on the same task learn representations with a shared subspace.  If every model learned a different representation, as the reviewer suggests, we would expect to see very high values for the A and B lines in Figure 1b.  However, we see values consistently less than 0.5, indicating that while representations are certainly not identical (subject to a linear transform), there certainly exists a shared subspace in the representations which captures much of the variance.
>
> >  I believe that the results are not strong enough to support the claims that are stated in the paper and the limited scope
>
> While our experiments do have a limit in their scope, we believe they do support the claims within this scope.  Extending the scope of our experiments provides and interesting avenue for future work.  We also emphasize that, to our knowledge, our work is the first to perform this type of analysis in the context of reinforcement learning in realistic environments.
>
> > Second, the authors claim in Section 5 that the random splits have little impact on the learned representation, but in Section 6 claim that the spatially disjoint splits have a noticeable impact on the representations. Without any measure of the impact, looking at figures 1b and 3a, I'm not convinced that this distinction is so obvious. The situation is worse when looking at the figures in the appendix, namely figures A3 and A5. If someone were to swap these two figures, my untrained eye would not be able to tell the difference.
>
>
> The differences in figures 1b and 3a may seem minor in scale, but they do have real impact.  The effect of this difference is pronounced on the transfer experiments (figure 2 vs. figure 3 (right)).
>
> > Some figure legends/captions can also be improved to include more information, such as explaining what exactly the shaded regions represent (I'm guessing one standard deviation from the mean over some unknown replicates).
>
> We will improve legends/captions.  The shaded regions represent a bootstrapped 95% confidence interval over 5 replicates.
>
> > or in Fig 3b making clear whether "A -> B" is done with "New Policy" or "Fine-tuned Policy". I am also curious as to why only one of these scenarios was experimented with in sections 6 and 7.
>
> Fig 3b is done with a New Policy.  We did not continue to examine the “Fine-tuned Policy” setting in latter experiments as Fig 2 showed that general navigation skills along with visual representations can be transferred between the tasks.  However, our PWCCA experiments only examine the visual encoder, thus utilizing the “Fine-tuned Policy” to ground PWCCA would introduce an un-accounted for variable.  We will clarify these points in future revisions.
>
>
> > Typos
>
> Thank you for pointing these out!  We will fix them in future revisions.

---

### Official Review · AnonReviewer4 · 2019-11-03
**Official Blind Review #4**

**Rating:** 3

**Review:**

This paper tries to analyze the similarities and transferring abilities of learned visual representations for embodied navigation tasks. It uses PWCCA to measure the similarity.  There are some interesting observations by smart experimental designing.

I have several concerns.

- for the non-disjoint experiments, the difference between A and B is that the subsets contain different instances. The objects in subsets A and B may have the same category. The objects with the same category may share similar surrounding environment. Thus, the visual inputs for the training model on A and B may just have minor differences. This point is also related to the spatial coverage used in the paper. Since the visual input is similar, why is the conclusion in Figure1(b) non-trivial?

- for the transferring experiments, in the beginning, the finetuning way is better than the new training makes sense. But, why do the results of learning a new policy from scratch will inferior to the finetuning way when training to convergence? The two experiments are both performed on the same fixed visual encoder.

- I think the experiments can not support the argument that residual connections help networks learn more similar representations. Will other structures such as VGG also learn similar representations? Will the degrees of similar representations be proportional to the accuracy of the classification tasks and the modified residual network still outperforms the squeezenet? The more straightforward ablation studies might be that we remove all shortcuts of the ResNet as the plain version.

=========================================================
After Rebuttal:

I thank the author for the response. I still think the evaluations and experimental settings cannot fully support the conclusions. So I keep the original score.

I hope the comments are useful for preparing a future version of this work.

**Experience Assessment:**

I do not know much about this area.

**Review Assessment: Checking Correctness Of Derivations And Theory:**

N/A

**Review Assessment: Checking Correctness Of Experiments:**

I carefully checked the experiments.

**Review Assessment: Thoroughness In Paper Reading:**

I read the paper thoroughly.

---

> ### Author Response · Authors · 2019-11-15
> **Response to R4**
>
> >  for the non-disjoint experiments, the difference between A and B is that the subsets contain different instances. The objects in subsets A and B may have the same category. The objects with the same category may share similar surrounding environment. Thus, the visual inputs for the training model on A and B may just have minor differences. This point is also related to the spatial coverage used in the paper. Since the visual input is similar, why is the conclusion in Figure1(b) non-trivial?
>
>
>
> In the case of the non-disjoint experiments, the models trained on A and models trained on B do see very similar data (even utilizing objects that only have one unique instance, A and B will still cover the entire environment in the majority of random splits).  While the models trained on A and B do see very similar inputs, they are trained (from scratch) to perform similar but disjoint tasks.  The conclusion drawn in Figure 1(b) is non-trivial as one would expect that the model would learn a visual representation that is dependent on its task. [1] performs an experiment where one set of networks learn the true label on CIFAR10 images while another set learns a random label for each image and find that the two sets of networks learn different representations for the same set of images.
>
> Furthermore, the ability of reinforcement learning to overfit (to even arbitrarily complex tasks and environments) [2] does imply that it is capable of learning representations that are highly tuned to the environment and task.
>
>
> > for the transferring experiments, in the beginning, the finetuning way is better than the new training makes sense. But, why do the results of learning a new policy from scratch will inferior to the finetuning way when training to convergence? The two experiments are both performed on the same fixed visual encoder.
>
> The results in all settings will eventually converge to the same policy (in the limit).  We did not train to convergence as our motivation was to demonstrate sample efficiency.
>
>
> > The more straightforward ablation studies might be that we remove all shortcuts of the ResNet as the plain version.
>
> This is a great suggestion!  We will include this experiment in the final version of the paper.
>
> >  I think the experiments can not support the argument that residual connections help networks learn more similar representations. Will other structures such as VGG also learn similar representations? Will the degrees of similar representations be proportional to the accuracy of the classification tasks and the modified residual network still outperforms the squeezenet?
>
> We modified our ResNet35 network to have the same number of parameters as SqueezeNet, ResNet35 and SqueezeNet have a comparable number of architectural “blocks” (9 vs. 12), and both networks achieve comparable performance on all tasks.  Thus the only difference between the two networks is the architectural “blocks” used (SqueezeNet Fire blocks vs. ResNet Bottleneck blocks) and the aforementioned skip connections.
>
> [1] Morcos, Ari, Maithra Raghu, and Samy Bengio. "Insights on representational similarity in neural networks with canonical correlation." Advances in Neural Information Processing Systems. 2018.
>
> [2] Zhang, C., Vinyals, O., Munos, R. and Bengio, S., 2018. A study on overfitting in deep reinforcement learning. arXiv preprint arXiv:1804.06893.

---

### Decision · Program_Chairs · 2019-12-19

**Decision:**

Reject

**Comment:**

The general consensus amongst the reviewers is that this paper is not quite ready for publication, and needs to dig a little deeper in some areas.  Some reviewers thought the contributions are unclear, or unsupported.  I hope these reviews will help you as you work towards finding a home for this work.